# Interface between Bats and Pigs in Heavy Pig Production

**DOI:** 10.3390/v13010004

**Published:** 2020-12-22

**Authors:** Stefania Leopardi, Pamela Priori, Barbara Zecchin, Gianpiero Zamperin, Adelaide Milani, Francesco Tonon, Mirco Giorgiutti, Maria Serena Beato, Paola De Benedictis

**Affiliations:** 1Laboratory for Viral Emerging Zoonoses, Istituto Zooprofilattico Sperimentale delle Venezie, Viale dell’Università 10, 35020 Legnaro PD, Italy; bazecchin@izsvenezie.it (B.Z.); pdebenedictis@izsvenezie.it (P.D.B.); 2OIE Collaborating Centre for Diseases at the Animal/Human Interface, Istituto Zooprofilattico Sperimentale delle Venezie (IZSVe), Viale dell’Università 10, 35020 Legnaro PD, Italy; gzamperin@izsvenezie.it (G.Z.); amilani@izsvenezie.it (A.M.); 3Cooperativa S.T.E.R.N.A., Ecological STudies Research Nature Environment, Via Pedriali 12, 47100 Forlì, Italy; pamela.priori@gmail.com; 4Laboratory for Viral Genomics and trascriptomics, Istituto Zooprofilattico Sperimentale delle Venezie (IZSVe), 35020 Legnaro PD, Italy; 5Suivet, Via Ernesto Che Guevara 55, 42123 Reggio Emilia, Italy; tonon@suivet.it; 6Freelance Veterinarian, 33010 Pagnacco UD, Italy; Tevdem@libero.it; 7Laboratory for Diagnostic Virology, Department of Animal Health, Istituto Zooprofilattico Sperimentale delle Venezie (IZSVe), 35020 Legnaro PD, Italy; msbeato@izsvenezie.it

**Keywords:** bats, pigs, spillover, viruses, reassortment

## Abstract

Bats are often claimed to be a major source for future viral epidemics, as they are associated with several viruses with zoonotic potential. Here we describe the presence and biodiversity of bats associated with intensive pig farms devoted to the production of heavy pigs in northern Italy. Since chiropters or signs of their presence were not found within animal shelters in our study area, we suggest that fecal viruses with high environmental resistance have the highest likelihood for spillover through indirect transmission. In turn, we investigated the circulation of mammalian orthoreoviruses (MRVs), coronaviruses (CoVs) and astroviruses (AstVs) in pigs and bats sharing the same environment. Results of our preliminary study did not show any bat virus in pigs suggesting that spillover from these animals is rare. However, several AstVs, CoVs and MRVs circulated undetected in pigs. Among those, one MRV was a reassortant strain carrying viral genes likely acquired from bats. On the other hand, we found a swine AstV and a MRV strain carrying swine genes in bat guano, indicating that viral exchange at the bat–pig interface might occur more frequently from pigs to bats rather than the other way around. Considering the indoor farming system as the most common system in the European Union (EU), preventive measures should focus on biosecurity rather than displacement of bats, which are protected throughout the EU and provide critical ecosystem services for rural settings.

## 1. Introduction

The number of emerging zoonotic and epizootic pathogens has proven to be increasing in recent years, especially from the wildlife reservoir [1]. Several authors are suggesting that humans themselves are the most powerful drivers of viral spillover in the modern era, known as the Anthropocene [1,2,3]. Indeed, the exponential growth of the human population is inadvertently affecting global ecosystems and changing historical balances between species, including equilibria existing between pathogens and their natural hosts. While the epidemiological thread of most emerging diseases is linked to the hunt, trade and butchering of wild animals that is a common practice in many Asian and African communities [3,4], it is likely that other anthropogenic activities are more relevant in the Western hemisphere. Among these, the industrialization of farming driven by the increasing demand for animal protein is certainly triggering critical changes in the relationship between domestic animals and wildlife, recognized as the main source of modern epidemics [1,5]. Indeed, livestock production has expanded to use increasing amounts of land, stealing natural habitats from wild species but providing at the same time new, abundant and reliable opportunities of shelter and forage. In turn, the overlap between wildlife and domestic animals is strongly increasing and wild species are subject to significant changes in population dynamics, patterns of movement and nutritional balance, possibly leading to enhanced shedding of the pathogens that are associated with them [2,6,7,8,9,10,11]. In terms of disease emergence, this translates into a higher likelihood for wildlife viruses to spill across this closer interface between wild and domestic animals, posing a risk in terms of animal health and potentially favoring future zoonotic transmission [5,7,12,13]. Among livestock, fast-growing species such as pigs are experiencing a major growth in recent times [14], providing the optimal scenario for the emergence of wildlife pathogens. Indeed, the pig industry was claimed to be responsible for the pandemic of H1N1 influenza A in 2009 and for the Malaysian epidemic of Nipah virus [15,16,17]. Among wildlife, bats have been particularly studied as natural hosts for viruses with zoonotic and epizootic potential [18]. Indeed, several of the viruses responsible for modern epidemics in both humans and animals have been linked with these animals from an evolutionary point of view, including the newly emerged SARS-CoV-2, Nipah, Marburg and Ebola virus but also animal pathogens such as the swine acute diarrhea syndrome coronavirus (SADS-CoV), which has caused the death of almost 25,000 piglets across Chinese farms [19,20,21]. While all these epidemics originated in Asia or Africa, viruses shown to be a potential risk to animal and human health have been confirmed to widely circulate in European bats as well, including lyssaviruses, filoviruses, coronaviruses, astroviruses, reoviruses, paramyxoviruses, bunyaviruses and hantaviruses [22,23]. However, direct human exposure to bats is predicted to be much lower in Europe, where big bat colonies are usually protected from people disturbance, and risky activities such as animal hunting and bush meat consumption are not practiced nor allowed. In this context, viral emergence might be favored by passages in domestic animals, which could provide an epidemiological link between bats and humans and even amplify the virus.

In this study we investigated viruses at the interface between bats and pigs in Italy. As the transmission of a pathogen requires first and foremost the contact between natural and recipient hosts, the main objective of our research was to investigate the level of bat biodiversity and to quantify and characterize the activity of each species in intensive piggeries of northern Italy. In addition, we aimed to perform a preliminary investigation on enteric viruses found in these hosts that proved able to cross this interface, assuming indirect transmission through fecal contamination as the main pathway for the inter-species viral transmission in indoor productive systems. For this objective, we selected astroviruses (AstV), coronaviruses (CoVs) and mammalian orthoreovirus (MRV), as they display a broad host range, include several species/strains infecting pigs and are highly diversified and frequently found in European bats [24,25,26,27,28,29,30]. In addition, these viruses are all excreted through the feces of their hosts, with indirect transmission further favored by their high stability in the environment, which enlarges the temporal window for cross-species transmission [25,29].

## 2. Materials and Methods

We investigated the relationship between bats and pigs in heavy pig production and we performed a preliminary analysis on viruses circulating at this interface. For this study, we focused on Italian intensive farms located in the Friuli Venezia Giulia and Veneto regions, accounting for the far northeastern Italian territory (Appendix A). This area is characterized by a low to medium density of pig industries, most of which are devoted to the growth of heavy pigs for the production of traditional dry-cured ham, with particular reference to San Daniele ham of protected origin (DOP). In addition, around one third of farms host the reproduction of purebred and hybrid breeds admitted by the ham consortia, using mostly an open cycle where parents might be obtained from outside the farm [31]. For our study, we selected four intensive farms showing preliminary evidence of the circulation of bats, as identified through 30 min of bioacoustics recording in the surrounding area. Similarly, we assessed the presence of different bat species and their activity within the perimeter of piggeries using bioacoustic non-invasive methods.

### 2.1. Ecological Interface between Bats and Pigs

Basically, bioacoustic monitoring consists in the recording and analysis of the echolocation calls that all European bat species produce for orientation and prey detection, consisting in sonar signals generated in the larynx and emitted through either the mouth or the nostrils [32]. For this study, bat sounds were recorded in each pig farm using an automatic bat recorder (D500X; Pettersson Elektronik AB, Uppsala), an average of 12.4 days (min 10–max 15), depending on the activity level of the bats which affects the charging time of the batteries.

The machine was programmed to record all bat sounds at four-hour intervals between sunset and midnight, the first peak of activity, when all species leave their shelters and go drinking before dedicating to feeding and exploration. Hereafter, we refer to this interval of time using the term “night”.

Recordings were analyzed with BatSound 4.12 (Pettersson Elektronik AB, Uppsala) and Raven Pro 1.5, with no automated algorithms. All echolocation calls referred to bats were manually selected and the sonograms of each sequence were analyzed to determine the bat species combining results from visual inspection, measurements of at least 11 parameters relating to frequency (KHz) intensity (dB), emitting time (ms) and comparison with a parameter database. Calls which did not achieve sufficient classification performances were associated with the host genus only, with particular reference to *Nyctalus* spp. and *Myotis* spp. Once all sonograms were referred to their specific host, they were classified according to their use in: echolocations calls, used for orientation during flight and for environmental scanning; feeding buzzes, sequences of very short echolocation calls emitted at extremely fast repetition rate as bats detect and approach preys; and social calls, characterized by complex time-varying modulations of amplitude and frequency in audible range that are specific for each bat species [32]. We used these data to quantify the presence of bats and to characterize their habits within farms, by calculating basic parameters used in bioacoustics that include species richness and relative occurrence of each species of bat’s activity, and occurrence of feeding and social activity, determined for all bats and for each bat species. Accordingly, we characterized the activity of bats within each farm as simple passage, foraging or roosting, as detailed in the supplemental material (Appendix A).

In addition to bioacoustics analyses, we evaluated the suitability of each farm to host individual bats or bat colonies by assessing their level of disturbance, night illumination, sun exposure, ventilation, and chances for shelter. We inspected all crevices, holes and gutters with further support of a thermal imager (FLIR E40 BX) and we searched the environment for indirect signs of bat activity, including the presence of guano and marks of bat-specific predation. We also screened the surrounding area for bat colonies in potential roosts, focusing on abandoned buildings within a range of 5 km.

### 2.2. Presence of Viruses in Bats and Swines from Piggeries

Viruses were investigated in pigs and bats from piggeries, focusing on enteric viruses characterized by medium-high environmental resistance that can be easily transmitted even in the absence of direct contact between bats and pigs, which is predicted to be limited in Europe where pigs are mostly reared indoor. The collection of bat fecal samples was performed in July 2018, during the fieldwork activities described above. Samples were collected in pools below bat roosts that were identified in two piggeries. On the other hand, we collected pooled swine samples from animal boxes in September 2018, sampling different sheds from all stages of production (if present) and always including the sheds closer to the sewage tanks and the bat detector. All fecal samples were stored at −80 °C soon after field collection until laboratory analyses.

Fecal pellets from bats and pigs were homogenated at 1:10 dilution with antibioted phosphate buffered saline (PBS-A) before base extraction using QIAamp Viral RNA Mini kit (QIAGEN, Hilden, Germany). Based on the kit handbook, we assumed coupled extraction of DNA and RNA, allowing for the genetic identification of bats and the amplification of viral RNA. We thus confirmed the host species by sequencing the cytochrome oxidases I (COI), using primers modified from [33] (available upon request). We screened all samples for mammalian orthoreoviruses (MRVs), coronaviruses (CoVs) and astroviruses (AstVs) using broad-spectrum protocols of reverse transcription polymerase chain reaction (RT-PCR) directed towards highly conserved genes, including the L1 segment for MRV [34] and the RNA-dependent RNA polymerases (*RdRp*) for the genera *Alpha* and *Betacoronavirus* [35] and *Mamamstrovirus* [36]. Selected positive samples for each virus were further characterized, including sequencing of the partial S1 segment of MRVs using available primers and 3′ elongation Cov’s *RdRp*, using in-house primer pairs (Appendix A) [28,37]. In particular, sequences of CoV’s *RdRp* longer than 916 base pairs (bp) can be classified under the *RdRp* group units (RGU) [37]. Whole genome sequencing (WGS) was undertaken for MRVs isolated in VERO cells (one from bats and one from pigs), as detailed elsewhere [38].

### 2.3. Phylogenetic Analyses

Viral sequences obtained in the study were aligned with reference strains using the G-INS-1 and default parameters implemented in Mafft [39]. For CoVs, AstVs, and for the sole S1 and L1 segments of MRV, original sequences obtained in the study were aligned with reference strains from CoV species from bat, swine, human, rodents and other micromammals. In addition, we included bat strains genetically related with swine or human viruses, and all unclassified strains from European bat species found within piggeries during the study. Final alignments included 3402 bp for MRV’s L1, 1484 bp for MRV’s S1, 1225 bp for CoV’s *RdRp* and 1632 bp for AstV’s *RdRp*. Additionally, the complete coding region of MRVs was further aligned based on the same set of reference strains for the 10 segments, including our original sequences, prototype strains of the different serotypes, plus all viruses showing the highest BLAST identity with our new sequences in one of the 10 segments and for which the complete genome was available. MEGA 6 was then used to determine pairwise nucleotidic and amminoacidic distances using the p-distance approach [40].

Maximum likelihood (ML) nucleotide phylogenetic trees were inferred using PhyML (version 3.0), employing the GTR+Г4 substitution model, a heuristic SPR branch-swapping algorithm and 1000 bootstrap replicates [41]; obtained trees were edited online for graphical display using iTOL [42].

As for the whole genomes of MRV, we developed an algorithm for the visualization of phylogenetic incongruence to screen for events of gene reassortment, as detailed in the supplemental material (Appendix A). Briefly, we aligned ML trees referred for each segment of the same set of sequences, connecting equivalent sequences across trees with the number of crosses reduced to the minimum.

## 3. Results

We investigated four pig farms for the production of heavy pigs with a potential size ranging between 3000 and 5000 individuals; three of them were located in the region of Friuli Venezia-Giulia and the remaining one was in Veneto (Appendix A). Only one farm was devoted to the fattening of pigs (farm 4), while the others were reproduction stations with an open cycle. Interestingly, farm 4 declared to sell reproductive pigs to fattening farms, including farm 2 of this study. All farms reared pigs exclusively indoors, and had animal houses characterized by very high ventilation and light, likely preventing bats from entering. Accordingly, we did not find any signs of bat presence (i.e., feces and signs of predation) inside the shelters. On the other hand, all farms were located within rural areas in close proximity to water, thus suggesting they might be suitable areas for bats. Indeed, bioacoustics analyses showed a high level of both bat activity and bat biodiversity around all pig farms. In particular, we obtained acoustic evidence for the circulation of at least 13 bat species out of the 27 confirmed in the area (Table 1). In general, acoustic data suggest that half of the species is likely used to foraging within pig farms, while the remaining might just occasionally pass through. The most frequent and widely present species were *Hypsugo savii, Pipistrellus kuhlii* and *Pipistrellus pipistrellus* that showed a mean activity higher than eight passes per night (Table 2). Although the lack of social calls and feeding buzzes may suggest that these bats might only pass through some of the farms, our data support the possibility that they can also roost within the perimeters of pig industries. Among the other species, *Eptesicus serotinus* was detected in 2/4 farms, that, according to our data, likely represent habitual foraging areas and might also support animal roosting (Table 2). Similarly, the recording of bat calls early after sunset confirmed roosting in the proximities of farms for the other six species (Table 2). Regarding actual roosting within the perimeter of the farms, the conformation of gutters made them suitable roosts in three out of four piggeries, among which farm 2 actually hosted a maternal colony of around eight to 10 individuals of *P. kuhlii* at the time of sampling. In addition, most farms presented broken bricks and wall crevices, also considered suitable to host bats, with a pile of guano detected in the feed storage of farm 1. In addition, few fecal pellets were found in most piggeries above roofs, underneath gutters and within window blinds, except for farm 3 that was considered unsuitable to shelter bats due to its highly modern structure. In two out of four cases, ruins and abandoned buildings were located in the proximity of the farm, but we detected no colonies or individual bats during fieldwork.

In order to carry out our viral investigation, we collected bat fecal samples from two out of four farms, where the presence of bat roosts allowed us to collect fresh environmental samples, for a total of two and 16 pools, respectively, in farms 1 and 2. Genetic data identified the presence of *P. kuhlii* and *Myotis nattereri* in spots identified from farm 2 and 1, respectively. On the other hand, we analyzed 65 samples from swine, collected later in the season thus hampering any cross-contamination between the two hosts.

We found all the investigated viruses, namely astroviruses, coronaviruses and mammalian orthoreoviruses, in both bats and pigs from three farms. Samples from farm 3, accounting for the smaller data about bat occupancy, were all negative. No relevant clinical symptoms were recorded from swine during sampling.

Regarding CoVs, one and seven samples were positive in bats and pigs, respectively. All strains were classified based on partial sequences of *RdRp* and highly resembled to known species already described in the same host species. In detail, all CoVs found in pigs were associated with known swine CoVs, namely porcine haemagglutinating encephalomyelitis virus (PHEV) and porcine epidemic diarrhea virus (PEDV). On the other hand, the pipistrelle-*Alphacoronavirus* from farm 2 showed the highest nucleotide identity of 98.2% over 1225 bp of the *RdRp* with CoVs found in Italy in 2010 from the same bat species, confirming its classification under the same RGU (Figure 1). In pigs, the percentage of positivity ranged from 9.7% to 12.5% for PHEV, which was found in three farms, while PEDV was detected only from farm 2 in 10% of individuals tested (Figure 1). Co-circulation of PEDV and PHEV was confirmed at similar rates in farm 2, where an overall 20% of positivity was recorded [43]. Partial *RdRp* sequences of CoVs were identical between farms 4 and 2, which were selling animals to each another, while they differed slightly between Friuli Venezia Giulia and Veneto, showing 1.8% nucleotidic and 0.8% amminoacidic distance (Figure 2).

Similar results were obtained for astroviruses, with identification of four strains in pigs belonging to three species of swine astroviruses and a single strain in pipistrelle bats from farm 2. The percentage of positivity in pigs ranged from 3.2% to 12.5% in positive farms, with two strains co-circulating at similar rates in farm 4. Astroviruses found in farms 2 and 4 clustered together in the phylogenetic tree (Figure 3). Interestingly, the astrovirus found in bats (farm 2) was not typical of this mammalian host but was rather related with swine species (Figure 1), showing the highest nucleotidic identity of 94.1% with a pig strain from farm 1 located in the Veneto Region (Figure 3), which reached 100% identity at the amminoacidic level.

Finally, MRV was the most frequent virus found in swine, showing an average of 37% (0–65) of positivity for a total of 12 strains identified. In addition, a single strain was identified in bats from farm 1. As for CoVs and AstVs, MRVs found in pigs were consistently correlated with swine strains and largely clustered upon sampling location for both segments L1 and S1 (Figure 4). Within-farm mean distance was 0.6% and 2% for L1 and S1 respectively, while divergence between strains from different farms was 5.9% in mean for L1, while it ranged from 20% to a complete switch in serotype regarding S1. Indeed, serotype 3 circulated in two out of three positive farms, while serotype 2 was established only in farm 1 (Figure 4A). Interestingly, MRVs from fattening farm 2 and the relative reproduction farm 4 formed sister clades showing 99.6% mean nucleotide identity based on L1 phylogeny (Figure 4B), while being only distantly related (83% mean nucleotide identity) based on S1 (Figure 4A). Regarding the single MRV found from the bat species *M. nattereri*, L1 and S1 resembled bat strains the most, with 96.9% and 91% nucleotide identity with isolates from *M. myotis* and *P. kuhlii*, respectively (Figure 1 and Figure 4). Further data obtained for the whole genome sequencing of this variant, named 18RS2900–2/*M. nattereri*/Italy/2018, showed greater divergence of at least 3.1% from other strains published in public databases in all other segments, with higher identity with MRV from various non-flying mammals, including swine (M2, S3 and S4), chamois (L2 and L3), humans (M3) and civets (S2) (Figure 1). Phylogenetically, we found no evidence for a direct progenitor for segments L3, M2, M3 and S3 and no reads for M1 (Figure 1). In addition, we found no correlation between this bat strain and MRVs in pigs from the same farm either during this study or in previous years [38] as testified by mean nucleotide distance of 18% that spanned from 9.5% (S4 of the bat strain versus the swine strain of 2018) to 60.6% (S1 of the bat strain versus the swine strain of 2016) (Appendix A). On the other hand, the pig strain that was fully characterized from farm 1 (18DIA90178-3/swine/Italy/2018) grouped consistently with MRV3/16DIA52154-4/swine/Italy/2016, found in 2016 in the same pig population (Genbank accession numbers: MT151659-MT151678) in 7/10 segments (namely L1, L2, L3, S2, S3, M1, M3), while showing low identity and phylogenetic discordance for the remaining segments (namely S1, S4 and M2) (Figure 5), as discussed elsewhere [38]. While phylogenetic analyses supported a possible swine origin for the reassorted S1 and S4 segments, M2 of 18DIA90178-3/swine/Italy/2018 clustered with strains from voles and bats, showing nucleotidic identity of 89.4% with strain MRV_47Ma (GenBank accession number: KX384850), isolated in Hungary in 2006 from a common vole (*Microtus arvalis*) and 89% with strain SL-05 (GenBank accession number: MG457112), isolated in Slovenia from *M. myotis* (Figure 1 and Figure 5). Interestingly, 18DIA90178-3/swine/Italy/2018 clustered with these strains at the level of S1 as well, despite a higher identity being found for this gene with a swine MRV from Taiwan (Figure 1, Figure 4A and Figure 5).

## 4. Discussion

In this study, we combined expertise from ecology and virology to investigate the interface existing between bats and pigs in intensive production, including the description of pathogens that can be transmissible between these hosts and might pose a risk for either public or animal health.

We confirmed that bats are frequently found flying above Italian intensive pig farms with high biodiversity and activity. Indeed, we were able to record 15 species, accounting for more than half found in northeastern Italy (n: 27). These data are consistent with findings from habitats previously considered to be far more attractive for bats, such as woods or the Mediterranean scrub [44,45]. It is likely that such an unexpected result is related to the aggregation of insects around sewage tanks, a hypothesis that is supported by the intense forage activity up to 72% (mean 33%; range 14–72%) recorded in this study. In addition, bioacoustics data suggest that almost half of the species seem to roost close to or within the farms, supporting the idea that bats might also be attracted by the complexity of anthropogenic structures, which are able to provide suitable refuge for species with different ecological needs, as already suggested by Afelt and co-authors [46]. In general, pipistrelle bats (*P. kuhlii*, *P. pipistrellus* and *H. savii*) were the most frequent species in pig farms, in line with what is observed in most habitats in Italy. Indeed, these species have been experiencing a unique range expansion in the last decade thanks to their adaptability to different ecological niches and their flight plasticity, which allows them to exploit the majority of habitats, even if strongly anthropized [6]. However, we detected also rarer species from the genera *Myotis*, *Rhinolophus* and *Nyctalus*, not considered anthropophilic species, suggesting that the rural ecosystem is more important for bat conservation than previously realized.

As a consequence of this high biodiversity, pigs might be exposed to several bat viruses, including different species of CoVs and AstVs, that are referred to as species-specific in their natural hosts [29,47]. For example, 15 CoV species have been described in bat hosts recorded in this study (blue strains shown in Figure 2). However, we found no evidence of such pathogens among pigs, suggesting that spillover events are, as expected, rare and stochastic [7], and that a bigger sample size is needed for such an investigation. In addition, a serological approach might help to elucidate spillover events especially in case of low prevalence [48]. Interestingly, the whole genome of serotype 2 MRV circulating in farm 1 strongly supports the transmission of bat-associated genes to pigs, likely introduced in the herd between 2016 and 2018 through reassortment. Despite the fact that we were not able to spot the exact parental strains, two out of the three divergent segments (S1 and M2) clustered within the same group formed by viruses associated with bats (*M. myotis*) and rodents (*Microtus arvalis*), as described elsewhere [38]. Because MRVs show a higher frequency and diversity in bat species compared to rodents across Europe [28,49,50], and based on the evidence of high circulation of bats in piggeries, we suggest that chiropters are the most likely donor hosts for these genes. In this context, the greater mouse-eared bat *M. myotis* was found circulating within farm 1 through bioacoustics, even if no guano was collected. However, the possible role of rodents as primary hosts or either passive carriers within the facilities should not be underestimated, especially because it could provide an effective bridge between bats and the animal shelters [51]. Regardless of the source of parental segments, this event is particularly relevant in terms of animal and public health, because the large amount of genetic variation following virus reassortment has the potential to generate strains with increased transmissibility and/or pathogenicity [52,53]. Based on our data, it seems that by 2018 the reassortant MRV type 2 strain had largely replaced serotype 3 that was circulating in 2016, likely favored by the lack of herd immunity. We suggest that similar amplification of bat genes can increase chances for a subsequent transmission to a susceptible human population. As a matter of fact, it was the high shedding of Nipah virus from diseased pigs that determined the infection of animal caretakers, triggering the Malaysian epidemic in humans [54]. In this context, it is critical to notice how our bat isolate 18RS2900-2/*M. nattereri*/Italy/2018 carries a S1 segment showing 92% nucleotide identity with a strain associated with human neural infection [55], so that reassortment with this particular strain would have been of more concern in terms of public health.

Critically, all investigated viruses had circulated undetected in pig herds, likely because they were not associated with severe clinical manifestation. This fact is particularly worrisome considering that MERS-CoV had also silently circulated in livestock for more than 30 years before emerging as a severe human respiratory pathogen in 2012 [56]. Independent of the fact that this pathogen likely evolved from bats, dromedary camels turned out to be the main reservoir of MERS-CoV, which now makes the control of the disease in humans far more challenging as novel outbreaks are periodically fueled by animal to human transmission in endemic areas [57,58]. While some studies described the asymptomatic circulation of MRVs, AstVs and PHEV [59,60,61,62], the detection of highly pathogenic coronaviruses such as PEDV was completely unexpected, suggesting that surveillance in swine should be enhanced to investigate the epidemiology of swine viruses. As most enteric viruses in swine cause a similar clinical presentation, we suggest implementing broad-spectrum molecular techniques to avoid delayed or missed diagnosis that can result in the persistence, the amplification or even the spread of pathogens [43]. In addition, the use of second- and third-generation sequencing would have been critical to identify co-infections, in order to characterize the viral quasi-species in swine and the level of recombination. However, the low quality and limited number of samples collected during this study prevented us from successfully using this approach, leaving some questions still open for investigations. In our study, we consistently found phylogenetic clustering of swine viruses (or at least the consensus sequences) upon the sampling location, meaning that most of the transmission events occur within single farms. However, the consistent correlation of viruses from farms 2 and 4 highlights the risks of pathogen dispersal through movements of animals in open cycles of pig production.

A major finding of our study was also the detection of a swine astrovirus in bat guano. AstVs are considered species-specific viruses, as they generally show phylogenetic clustering according to the host species [29]. In turn, bats are expected to be infected with their own variants, that are well differentiated from AstVs infecting either swine or humans [30,63]. Unexpectedly, AstV found in bats in this study clustered with swine species, including Italian strains, supporting a spillover event from swine to bats. Although this finding should be considered as preliminary, as there is no evidence that the bat was actually infected rather than simply passing the virus through the gastrointestinal tract, the identification of two other swine AstVs in Chinese pipistrelle bats further supports the idea that bats might be exposed to pig pathogens through the feeding of coprophagist insects more than the other way around. Data obtained in this study from the whole genome sequencing of the MRV isolated from the guano of *Myotis nattereri* further supports this hypothesis. Indeed, this strain clustered with MRVs from myotis bats only in the L1 segment, while the others showed the highest identity with viruses from various mammalian species, including pigs, humans, ungulates and civets. It is possible that this constellation of genes is associated with the exposure of bats to diverse viruses through the indirect interaction with faecal materials from a wide variety of hosts during feeding. In terms of disease emergence, bats can then represent mixing vessels for genetic admixtures between viruses from different hosts, a process that is strongly favoured for MRVs by the fragmented nature of their genome. In addition, we should not overlook the possible impact for these endangered animals of cross-species transmission of pathogens that have evolved from different hosts. Even if it is the general opinion that bats do not get sick from infections, the arrival in the USA of a fungal pathogen (*Pseudogymnoascus destructans*) that commonly infects bats from the old world (being apathogenic), is now pushing several bat species of North America towards the edge [64,65]. In this context, the possible spill-back of SARS-CoV-2 from humans to bats is now a main concern following its massive spread and transmission in the human population [66,67].

## 5. Conclusions

In this study, we found that pigs are likely exposed to several bat viruses, because farms support the high biodiversity of these animals even in intensive systems. However, negative results from virological analyses suggest that transmission from bats is rather rare, likely because direct contact between hosts is limited in typical European settings where pigs are reared mostly indoors. Therefore, in this reality strict biosafety measures to avoid the contamination of fomites and animal feed with bat droppings would be effective in controlling the majority of bat viruses, known and unknown, including the control of rodents as possible carriers. On the other hand, bat biodiversity should be maintained and preserved and the displacement of bats from farms is highly discouraged, especially considering that some of the species found to live in this habitat are classified as vulnerable and endangered by the International Union for the Conservation of Nature [68]. Preservation of bat populations within farms is also important for maintaining the essential ecosystem services provided by these animals, above all, the biological control of insects harmful to the agricultural and livestock sectors [45,69,70]. For example, it is worth mentioning how from bat guano we accidentally amplified the genome of *Plutella xylostella*, an important pest of crucifers commonly used as livestock forage [71].

In addition, our data support the exposure, if not the infection, of bats to viruses found in faeces from livestock, suggesting animal farming as a crucial interface for viruses not only passing from bats to domestic animals and, in turn, to humans, but also to passing according to the same steps backwards.

## Figures and Tables

**Figure 1 viruses-13-00004-f001:**
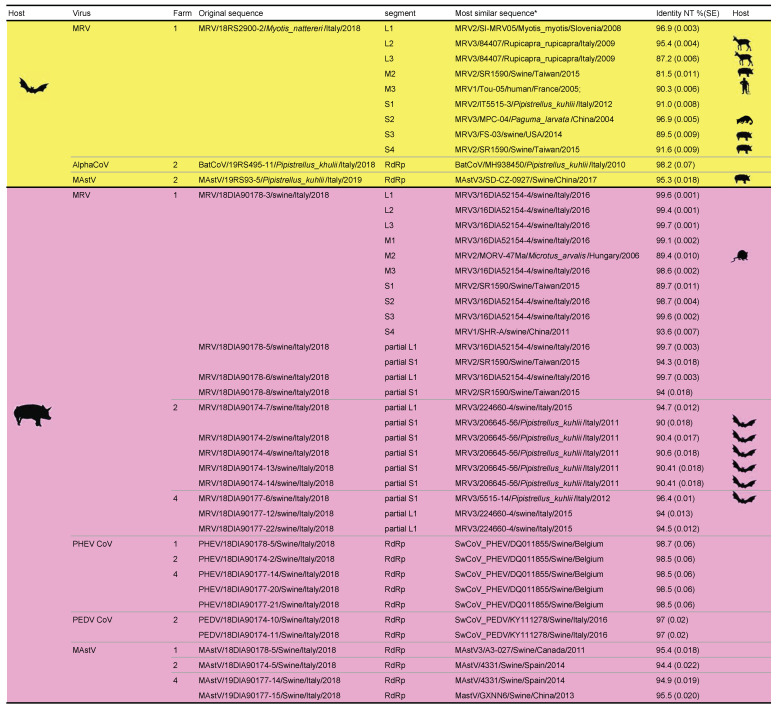
Sequences with the highest identity for all original strains found within the study, including the relative host species.

**Figure 2 viruses-13-00004-f002:**
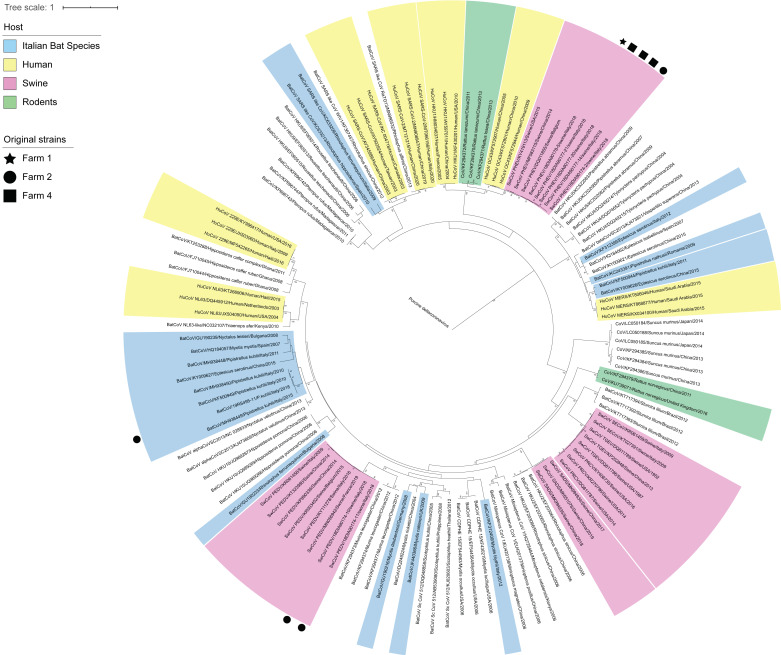
Phylogenetic tree based on the *RdRp* of alpha and betacoronaviruses. Sequences are coloured based on the host species, with original sequences marked according to the farm of detection.

**Figure 3 viruses-13-00004-f003:**
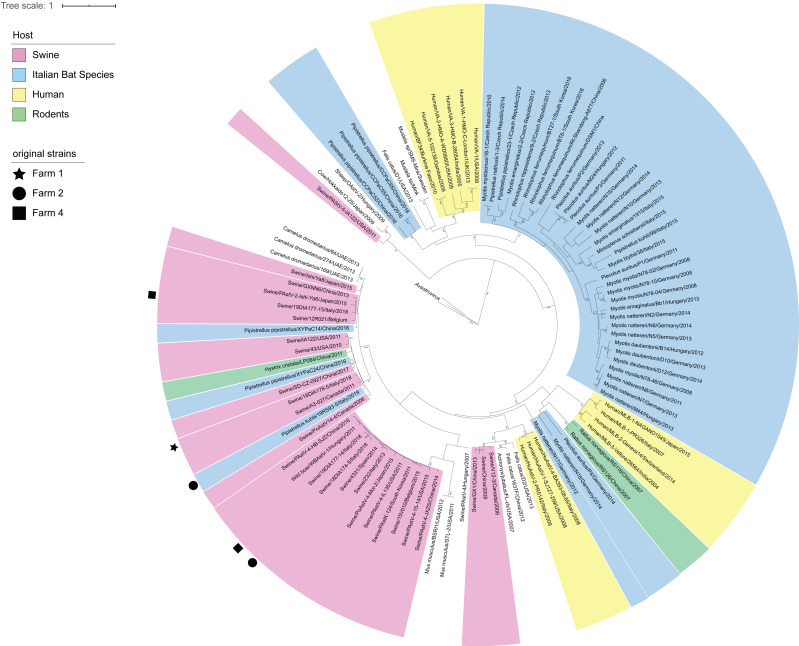
Phylogenetic tree based on the *RdRp* of mamastroviruses. Sequences are coloured based on the host species, with original sequences marked according to the farm of detection.

**Figure 4 viruses-13-00004-f004:**
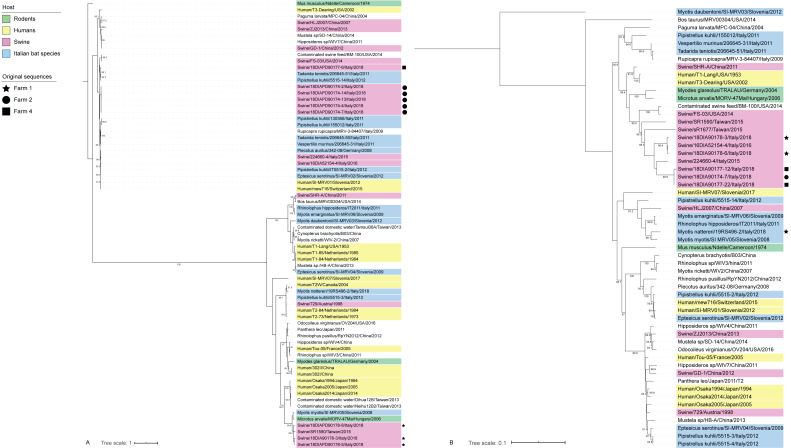
Phylogenetic tree of mammalian orthoreoviruses based on complete S1 (**A**) and L1 (**B**) segments. Bootstrap values >80 are not shown. Sequences are coloured based on the host species, with original sequences marked according to the farm of detection.

**Figure 5 viruses-13-00004-f005:**
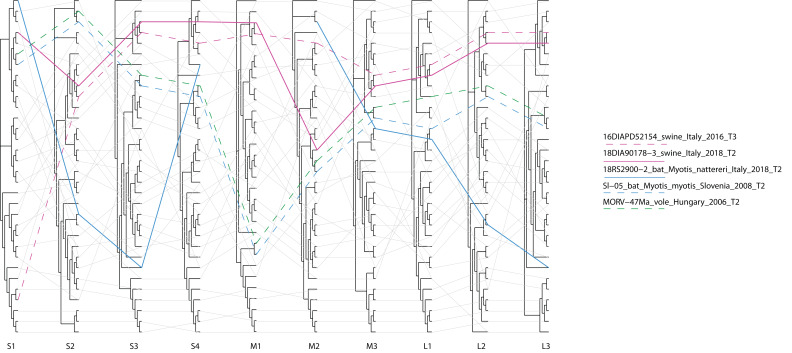
Graphical reassortment analysis using tanglegrams to explore the phylogenetic congruence between the 10 segments of MRVs. The figure shows the strong phylogenetic correlation across most of the genome between MRV strains found in pigs from farm 1, shown with full and dashed fuchsia lines, with incongruent phylogeny supporting reassortment in S1, S4 and M2 segments. The figure also indicates strains the most likely parental strains for S1 and M2 found in a myotis bat (blue dashed line) and a vole (green dashed line). Finally, the figure highlights how pig strains have different evolutionary origin than the MRV isolated from a myotis bat roosting in the same environment, shown as a full blue line. M1 is not shown as no reads were obtained from the bat’s MRV.

**Table 1 viruses-13-00004-t001:** Bat activity and biodiversity estimated using bioacoustics data.

Farm n.	Bat Activity (Passes/Night)	Feeding Activity (%)	Social Activity (%)	Species Richness (n)
1	142	17.6	4.2	9
2	67	28.4	4.5	9
3	83	72.3	22.9	5
4	93	14	1.1	6
	96.25 mean/farm	33.1 mean/farm	8.2 mean/farm	13 species in total

**Table 2 viruses-13-00004-t002:** Results of bioacoustics analysis per farm and for each bat species. Acronyms used to characterize the activity of bats: P: passage, scored as rare (rP), occasional (oP) or habitual (hP); F: foraging, score as occasional foraging (oF), foraging area (Fa) or habitual foraging area (hFa); R: roost, flagged as * if confirmed during inspection. See Appendix A for the assignation of categories.

Farm n.	Species	Activity (p/h)	Occurrence (%)	Feeding Activity (%)	Social Activity (%)	Presumed/Expected Activity
1	*H. savii*	14.8	21.13	0	0	hP; R
	*P. kuhlii*	16.6	23.24	54.6	15.2	hP; hFa; R
	*P. pipistrellus*	19.0	26.76	2.6	0	hP; oF; R
	*E. serotinus*	6.8	9.86	14.3	7.1	oP; hFa; R
	*N. leisleri*	8.5	11.97	23.5	0	oP; Fa; R
	*M. emarginatus*	0.5	0.70	0	0	rP; R
	*M. mystacinus*	0.2	0.70	0	0	rP
	*M. nattereri*	1.1	2.11	0	0	rP; R *
	*M. myotis/blythii*	0.2	3.52	0	0	rP
2	*H. savii*	4.6	17.39	83.3	16.7	oP; hFa
	*P. kuhlii*	5.4	20.29	21.4	0	oP; Fa; R *
	*P. pipistrellus*	6.5	24.64	23.5	0	oP; Fa
	*E. serotinus*	0.8	2.90	50	50	rP; hFa
	*N. leisleri*	0.2	13.04	11.1	0	oP; Fa
	*Nyctalus* sp.	0.2	1.45	0	0	rP
	*M. daubentonii*	0.9	2.90	0	0	rP
	*M. nattereri*	4.5	11.59	0	0	oP
	*R. ferrumequinum*	1.6	5.80	0	0	oP
3	*H. savii*	1.9	7.23	100	16.7	oP; hFa; R
	*P. kuhlii*	4.5	14.46	0	0	oP
	*P. pipistrellus*	3.7	69.88	93.1	31.0	hP; hFa; R
	*N. leisleri*	2	6.02	0	0	oP
	*R. hipposideros*	0.6	2.41	0	0	rP; R
4	*H. savii*	10.3	31.18	0	0	hP
	*P. kuhlii*	6.4	19.35	5.5	0	oP; oF
	*P. pipistrellus*	13.3	39.78	32.4	2.7	hP; hFa
	*T. teniotis*	0.8	2.15	0	0	rP
	*M. emarginatus*	1.9	6.45	0	0	oP
	*Myotis* sp.	0.3	1.08	0	0	rP
Distribution (% farms)		Mean/farm				
50	*E. serotinus*	3.8	6.4	32.1	28.6	P; F; R
100	*H. savii*	8	19.2	45.8	8.3	P; F; R
25	*M. daubentonii*	0.9	2.9	0	0	P
50	*M. emarginatus*	1.2	3.6	0	0	P; R
25	*M. myotis/blythii*	0.2	3.5	0	0	P
25	*M. mystacinus*	0.2	0.7	0	0	P
50	*M. nattereri*	2.8	6.9	0	0	P
75	*N. leisleri*	3.6	10.3	11.5	0	P; F; R
100	*P. kuhlii*	8.2	19.3	20.4	3.8	P; F; R
100	*P. pipistrellus*	10.6	40.3	37.9	8.4	P; F; R
25	*R. ferrumequinum*	1.6	5.8	0	0	P
25	*R. hipposideros*	0.6	2.4	0	0	P; R
25	*T. teniotis*	0.8	2.2	0	0	P

## Data Availability

Consensus sequences for all viruses were deposited in GenBank under accession numbers MW089336-MW089354, MW199190-MW199198 and MW385367-MW385371. MiSeq raw data were submitted to the NCBI Sequence Read Archive (SRA) under accession numbers SRR12704223, SRR12704224.

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
