# Peer review of "Interface between Bats and Pigs in Heavy Pig Production"

_viruses, 2020, doi:10.3390/v13010004_

Round 1

Reviewer 1 Report

This is an interesting and engaging article describing virus circulation in pigs and bats. This study provides valuable insight information into possible outcomes and virus evolution. I wish to admit that this study was interesting for me to read. After a well-focused Introduction, the Materials and Methods section provides detailed information on the approaches used. Sufficient information on data gathering is provided. All results are similarly well represented and accordingly illustrated. Data analyses were carried out. The Discussion summarizes well the work described above; it is interesting. Proper citations are provided to support the authors' statements. The interpretation of the data makes sense. Conclusions are consistent with the evidence and arguments presented. The authors properly address the main question posed.

To sum it up, the provided approach is valid for concluding and the whole study triggers significant interest. In my opinion, the manuscript by Leopardi et al. is devoid of major weaknesses.

Comments:
A) The writing has to be improved. I will bring below just a few out of many grammar mistakes:
Figure S1: Nnorth > North
Line: 124: And > and
Line 151: homogenate > homogenated
Table 2: Valori medi/azienda > Average per group
Line 240: highest > the highest
Suppl. material: nodes and leafs > nodes and branches
Suppl. material: c++ script > C++ code
B) Multiple phrases should be rephrased to improve readability (e.g. lines 129-131).
C) NGS approach (e.g. MiniION) is capable to provide more insights for the processed samples. If authors are unable to provide additional data, just reflect it in the discussion.

The aforementioned comments and remarks are not affecting the value of the results provided in the considered manuscript. Thus, I propose the publication of the current manuscript after thorough addressing of revealed issues.

Author Response

Dear reviewer, 

Many thanks for your positive evaluation of the manuscript. 

All the comments raised have been addressed accordingly. In particular:

A-B) The manuscript has been extensively edited by an English native speaker. Mistyping has been removed as well as some sentence re-written.

C)The authors acknowledge that NGS approach (e.g. MiniION) might have provided more insights into the occurrence of co-infections or even recombination events. However, the scarce amount and quality of samples collected during this study prevented us from successfully using this approach. We have therefore included a specific mention in the Discussion section. 

Reviewer 2 Report

Reviewer comments:

This is a generally well-written paper describing the relationship between bats and pigs in the heavy pig production and the preliminary analysis on viruses circulating at this interface. Your results are very interesting, in particular the detection of viruses (AstV and MRV) carrying swine genes in bat guano, indicating that bats can represent mixing vessels for genetic admixtures between viruses from different hosts. However, negative results from virological analyses of bat virus in pigs suggest that the strict biosafety measures are effective in controlling the transmission from bats to domestic animals and, in turn, to humans. The article agrees with the background and goals of the journal. I only have two minor remarks:

In Table 2: please verify the acronyms described in the legend with those in the table, i.e. aP and hFz.

Page 6 line 241: Table 3 is missing in the article.

Page 9 line 285: the Figure 3 concern astroviruses, not MRVs. Please verify.

In Supplemental material: Please verify the mistaken “Figure 3” in the first line of the chapter “Visualization of tanglegrams for the analysis of phylogenetic incongruence”

The authors in some references are incomplete, a revision is required.

Author Response

Dear reviewer, 

Many thanks for your positive evaluation of our study.

All the comments raised have been addressed in the revised form of the manuscript. In particular: 

Table 2: the acronyms described in the legend have been all revised.

Page 6 line 241: Table 3 is missing in the article. Addressed accordingly. 

Page 9 line 285: the Figure 3 concern astroviruses, not MRVs. Please verify. Addressed accordingly.

In Supplemental material: Please verify the mistaken “Figure 3” in the first line of the chapter “Visualization of tanglegrams for the analysis of phylogenetic incongruence”. Figure 3 was erroneusly referring to Figure 5.

References have been added accordingly.